# Biomarkers and Predictive Factors for Treatment Response to Tumor Necrosis Factor-α Inhibitors in Patients with Psoriasis

**DOI:** 10.3390/jcm12030974

**Published:** 2023-01-27

**Authors:** Teppei Hagino, Hidehisa Saeki, Naoko Kanda

**Affiliations:** 1Department of Dermatology, Nippon Medical School Chiba Hokusoh Hospital, Inzai 270-1694, Japan; 2Department of Dermatology, Nippon Medical School, Tokyo 113-8602, Japan

**Keywords:** psoriasis, tumor necrosis factor-α inhibitor, adalimumab, infliximab, certplizumab pegol, scalp, C-reactive protein, platelet to lymphocyte ratio

## Abstract

We performed a retrospective and observational study of patients with psoriasis. The aim of this study was to define the laboratory indicators reflecting the treatment response to tumor necrosis factor (TNF)-α inhibitors and the predictors for the treatment response. From January 2010 to June 2022, 28, 15 and 12 patients with psoriasis were treated with infliximab (IFX), adalimumab (ADA) and certolizumab pegol (CZP), respectively. The values of C-reactive protein (CRP), platelet to lymphocyte ratio (PLR), neutrophil to lymphocyte ratio and monocyte to lymphocyte ratio decreased in parallel with psoriasis area and severity index (PASI) at weeks 12 and 52 of treatment. The percentage reduction of the CRP was correlated with that of the PASI at week 52 in all patients and subgroups treated with IFX. The percentage reduction of the PLR was correlated with that of the PASI at week 52 in all patients. Linear multivariate regression analyses revealed that the presence of scalp lesions was associated with a high percentage reduction of the PASI at week 52 in the ADA subgroup. The CRP and PLR might act as biomarkers reflecting the treatment response to TNF-α inhibitors in patients with psoriasis. The presence of scalp lesions might be a predictive factor for a high treatment response to ADA in patients with psoriasis.

## 1. Introduction

Psoriasis is a chronic inflammatory skin disease with enhanced immune axis of tumor necrosis factor (TNF)-α/interleukin (IL)-23/IL-17 [1]. The biologics targeted at these cytokines have been advanced in recent years for the treatment of psoriasis. A total of 11 types of biologics (adalimumab (ADA), infliximab (IFX), certolizumab pegol (CZP), ixekizumab, secukinumab, brodalumab, bimekizumab, ustekinumab, guselkumab, risankizumab and tildrakizumab) are currently approved for psoriasis in Japan. Among these, TNF-α inhibitors, ADA and IFX have been used for a relatively long period, from January 2010, while another TNF-α inhibitor, CZP, was rather recently approved in December 2019 in Japan. Although these TNF-α inhibitors confer high efficacy against a rash, background factors of patients predicting treatment responses, such as age, sex, body mass index (BMI), disease duration, baseline psoriasis area and severity index (PASI), presence or absence of bio-switch or arthritis, have not been precisely investigated. Further, the laboratory indicators reflecting the treatment response to TNF-α inhibitors in psoriasis have not been established.

Psoriasis is also known to be a systemic inflammatory disease often affecting internal organs such as joints, circulatory organs, intestines or the liver. The C-reactive protein (CRP), the neutrophil-to-lymphocyte ratio (NLR), the monocyte-to-lymphocyte ratio (MLR) and the platelet-to-lymphocyte ratio (PLR) have been identified as indicators of systemic inflammation [2,3,4,5]. It is reported that higher baseline NLR and PLR levels in patients with rheumatoid arthritis were associated with non-response to TNF-α inhibitors at week 12 of treatment [6]. Recent studies revealed that NLR and PLR values in patients with psoriasis are higher than those in control healthy individuals [5,7]. Further, the values of the CRP, NLR and PLR were correlated with each other and decreased after treatment with ADA or IFX in patients with psoriasis [8,9,10]. It is thus indicated that these laboratory parameters might act as biomarkers (characteristics that are objectively measured and evaluated as indicators of normal biological processes, pathogenic processes or pharmacological responses to therapeutic intervention) [11] reflecting and/or predicting the treatment response to TNF-α inhibitors in patients with psoriasis.

We herein analyze the transition of the NLR, MLR, PLR or CRP during the treatment with TNF-α inhibitors, IFX, ADA and CZP in patients with psoriasis and tried to determine which of the parameters can act as biomarkers reflecting the treatment response. We also examined which of the patients’ background factors, including the baseline values of the above parameters, might predict a high treatment response to TNF-α inhibitors.

## 2. Materials and Methods

### 2.1. Study Design and Data Collection

A total of 55 Japanese patients with plaque-type psoriasis (≥18 years of age; 43 males and 12 females) living in Chiba Prefecture were treated with TNF-α inhibitors (IFX, ADA and CZP for 28, 15, 12 patients, respectively) from January 2010 to June 2022. Subcutaneous ADA was initially administered at a dose of 80 mg, then 40 mg at 2 weeks and 40 mg or 80 mg every 2 weeks thereafter. Intravenous IFX was initially administered at a dose of 5 mg/kg at 0, 2 and 6 weeks and every 8 weeks thereafter, and in some patients the dose was increased up to 10 mg/kg. Subcutaneous CZP was administered at 400 mg every 2 weeks. They were analyzed retrospectively using the medical record. This study was conducted based on the Declaration of Helsinki (2004) and was approved by the Ethics Committee of Nippon Medical School Chiba Hokusoh Hospital. The patients provided written informed consent.

The diagnosis of psoriasis for the patients was made clinically. Before treatment, we examined the patients’ age, body mass index (BMI), disease duration, presence or absence of bio-switch, past history of tuberculosis, arthritis, scalp, nail and genital lesions, diabetes mellitus, cardiovascular diseases or current smoking status.

The values of the PASI, NLR, MLR, PLR and CRP were analyzed at weeks 0, 12 and 52 of treatment. The proportion of patients who had achieved at least a 75% or 90% reduction in the PASI from the baseline (PASI 75 or PASI 90, respectively) or absolute values of the PASI ≦ 2 was calculated.

### 2.2. Statistical Analysis

All statistical analyses were performed using the EZR (Saitama Medical Center, Jichi Medical School). The Shapiro-Wilk test was used to assess the normality of the data distribution. Results are expressed as the mean ± standard deviation (SD) for variables with a normal distribution or as the median and interquartile range for variables with a non-parametric distribution. Differences between weeks 0, 12 and 52 were analyzed using repeated measures analysis of variance for normally distributed variables or using Friedman’s test for variables with non-parametric distribution. Bonferroni correction was used for post-hoc analysis. The differences between patients’ subgroups treated with IFX, ADA or CZP were analyzed using a one-way analysis of variance using the Tukey-Kramer test for normally distributed variables or using the Kruscal-Wallis with Steel-Dwass test for variables with non-parametric distribution. A comparison between two independent groups was performed using Student’s *t*-test for normally distributed variables or using the Mann-Whitney U test for variables with non-parametric distribution. Group comparisons for categorical variables were performed using Fisher’s exact test. The correlations of the variables with each other were analyzed by Spearman’s correlation coefficients. Statistical significance was set at *p* < 0.05.

A linear multivariate regression analysis was performed to determine the predictive factors for the high percentage reduction of the PASI at week 12 or 52 of treatment with TNF-α inhibitors. The analysis included only the variables with a *p* value < 0.05 in univariate analyses and was adjusted for age, sex and BMI. Variables with a variance inflation factor >10 were excluded to avoid multicollinearity.

## 3. Results

### 3.1. Background Factors of the Patients

Background factors of the patients are shown in Table 1, including the baseline values of the PASI, NLR, MLR, PLR and CRP before treatment. The incidence of bio-switch in the subgroups treated with ADA or CDP was higher than that of the IFX subgroup. The incidence of arthritis in the CZP subgroup (100%) was higher than that in the IFX subgroup. These results might reflect that IFX was the biologic approved for psoriasis for the first time in Japan and thus patients treated with IFX are mostly bio-naïve and that CZP is used as a first-line treatment for psoriatic arthritis (PSA) after approval in Japan due to its high accumulation rate in inflammatory joint lesions [12,13]. We then analyzed the relationship of the patients’ background factors with the baseline values of the NLR, MLR, PLR or CRP (Table 2). The baseline MLR was higher in patients with nail lesions than in those without. The baseline PLR was negatively correlated with age. The baseline NLR and PLR were negatively correlated with BMI. The baseline MLR was positively correlated with disease duration.

### 3.2. The Achievement Rate of PASI 75, PASI 90 and Absolute PASI ≤ 2 during the Treatment with TNF-α Inhibitors

We first evaluated the achievement rates of PASI 75, 90 and absolute PASI ≤ 2 during the treatment with individual TNF-α inhibitors in patients with psoriasis. During the treatment, until week 24, the magnitude and increasing speed of PASI 75 and 90 achievement rates appeared to be the highest in the IFX subgroup (70.4% of PASI 75 at week 24) followed by the ADA subgroup (53.3%), while those of the CZP subgroup (40%) appeared to be lower (Figure 1a,b). In contrast, the magnitude and increasing speed of achievement rate for absolute PASI ≤ 2 appeared to be the highest in the CZP subgroup (80% at week 24) followed by the ADA subgroup (66.7%), while those of the IFX subgroup (59.3%) appeared to be lower (Figure 1c). The achievement rates of PASI 75, 90 and PASI ≤ 2 in the IFX subgroup peaked at week 24, while those in the CZP subgroup gradually increased until week 52.

### 3.3. The Transition of PASI, NLR, MLR, PLR and CRP Values during the Treatment with TNF-α Inhibitors

In all patients and the IFX, ADA and CZP subgroups, the PASI scores at weeks 12 and 52 were significantly reduced compared to the baseline levels, while there were no significant differences between the PASI scores at weeks 52 versus 12 (Figure 2a). In all patients, the NLR, MLR, PLR and CRP values at weeks 12 and 52 were significantly reduced compared to the baseline levels, while there were no significant differences between these values at weeks 52 versus 12 (Figure 2b–e). The NLR values at week 12 in the IFX, ADA and CZP subgroups and those at week 52 in the IFX and CZP subgroups were significantly reduced compared to the baseline levels (Figure 2b). The MLR values at week 52 in the IFX subgroup were significantly reduced compared to the baseline levels, while not in the other subgroups (Figure 2c). The PLR values at week 12 in the ADA and CZP subgroups and those at week 52 in the IFX subgroup were significantly reduced compared to the baseline levels (Figure 2d). The reductions of the CRP at weeks 12 and 52 in the IFX, ADA and CZP subgroups were not statistically significant despite the significant reduction in all patients (Figure 2e).

### 3.4. Correlations between Individual Parameters

We next analyzed the correlations between individual parameters during the treatment with TNF-α inhibitors (Table 3). In all patients, the NLR, MLR, PLR and CRP were significantly positively correlated with each other at weeks 0, 12 and 52, except for no significant correlation between the PLR and CRP. In the IFX subgroup, the results were mostly the same as those in all patients, except for the significant correlation between the PLR and CRP at week 0 and no significant correlation between the MLR and CRP at week 12. In the ADA subgroup, significant correlations were seen between the NLR and PLR at weeks 0, 12 and 52 and between the NLR and CRP at weeks 0 and 12. In the CZP subgroup, significant correlations were seen between the NLR and MLR at week 52 and between the NLR and PLR at weeks 0 and 52. Thus, the values of the NLR, MLR, PLR and CRP were mostly correlated with each other before and after treatment with TNF-α inhibitors.

### 3.5. Correlations between Percent Reduction of PASI versus Those of NLR, MLR, PLR or CRP

We then analyzed if the percent reduction of the PASI in response to TNF-α inhibitors might be correlated with those of laboratory parameters. In all patients, the percentage reductions of the PLR and CRP were positively correlated with that of the PASI at week 52 (Table 4), indicating that the PLR and CRP might act as biomarkers reflecting treatment response to TNF-α inhibitors in psoriasis. In the IFX subgroup, the percentage reduction of the CRP was correlated with that of the PASI at week 52. In the ADA and CZP subgroups, there were no significant correlations between the percentage reduction of laboratory parameters and that of the PASI.

We then divided the patients into those who achieved PASI 75 (high responders) and those who did not (low responders) at week 12 or 52 and compared the percentage reductions of the NLR, MLR, PNR and CRP between the high and low responders (Table 5). The percentage reduction of the CRP was significantly higher in high responders than in low responders in all patients and in the ADA subgroup at week 52 and in the IFX subgroup at week 12. Thus, the percentage reduction of the CRP might discriminate between high and low responders in treatment with TNF-α inhibitors.

### 3.6. The Background Factors Predicting Treatment Response to TNF-α Inhibitors in Patients with Psoriasis

We then examined the background factors that can predict a high treatment response to TNF-α inhibitors, as evaluated using the percentage reduction of the PASI. We first examined whether age, BMI, disease duration and baseline values of the PASI or laboratory parameters affected the treatment response. In the ADA subgroup, the baseline PASI was positively correlated with the percentage reduction of the PASI at week 52 (Table 6). In the CZP subgroup, the baseline CRP was negatively correlated with the percentage reduction of the PASI at week 52 (Table 6). These results indicate that a high baseline PASI may predict a high treatment response to ADA at week 52, while a high baseline CRP may predict a poor response to CZP at week 52. There were no significant correlations between these variables versus the percentage reduction of the PASI at week 12 (Appendix A).

We next analyzed whether sex, presence or absence of bio-switch, arthritis, scalp, nail or genital lesions, diabetes, current smoking status or history of tuberculosis affected the treatment response to TNF-α inhibitors. Patients with genital lesions had a lower percentage reduction of the PASI at week 52 than those without, in all patients and in the IFX subgroup (Table 7). Patients with scalp lesions had a higher percentage reduction of the PASI at week 52 than those without in the ADA subgroup (Table 7). There were no significant associations between these variables versus the percentage reduction of the PASI at week 12 (Appendix A).

These results indicate that genital lesions may predict a poor response to IFX, while scalp lesions may predict a good response to ADA at week 52. There were no significant associations between the percentage reduction of the PASI at week 52 versus the presence or absence of nail lesions, diabetes, current smoking status or bio-switch (Table 8). There were no significant associations between these variables versus the percentage reduction of the PASI at week 12 (Appendix A). There was only one patient who had cardiovascular disease, and thus the association of the disease with the treatment response could not be evaluated.

Then linear multivariate regression analysis was performed to determine the predictive factors for the treatment response to TNF-α inhibitors, as evaluated using the percentage reduction of the PASI at week 52 (Table 9). The presence of scalp lesions was associated with a high percentage reduction of the PASI at week 52 in the ADA subgroup (Table 9). The results indicate that scalp lesions may be a predictive factor for a good treatment response to ADA at week 52 in patients with psoriasis.

## 4. Discussion

In this study, the values of the CRP, NLR, MLR and PLR as well as the PASI significantly decreased in response to treatment with TNF-α inhibitors. Further, the percentage reductions of the CRP and PLR were correlated with that of the PASI at week 52 in all patients. The patients who achieved PASI 75 at week 52 showed a higher percentage reduction of the CRP compared to those who did not, in all patients and in the ADA subgroup. These results indicate that the CRP and PLR might act as biomarkers reflecting treatment response to TNF-α inhibitors in psoriasis. In particular, the high percentage reduction of the CRP might indicate a high treatment response to TNF-α inhibitors. Tumor necrosis factor-α enhances the secretion of IL-6, which stimulates the production of CRP in hepatocytes [14]. Asahina et al. also reported that treatment with IFX and ADA reduced the levels of the CRP in patients with psoriasis [9]. The reduction of the CRP values by TNF-α inhibitors might be caused by their direct inhibition of TNF-α-induced synthesis of CRP [8]. Though the reduction of the CRP values at weeks 12 and 52 after treatment was statistically significant in all patients, it was not significant in the individual IFX, ADA and CZP subgroups (Figure 2e). This is possibly due to the small sample size of each subgroup and/or the usage of a conventional method to measure CRP. The high-sensitivity CRP method that can detect CRP values of less than 0.02 mg/dL might be preferable in low levels of the CRP [15].

Monitoring the percentage reduction of the PLR during treatment with TNF-α inhibitors might complement that of the CRP to evaluate the treatment response. Tumor necrosis factor-α induces megakaryocytic differentiation, promoting platelet production, contributing to the increase of the PLR [16]. Thus, TNF-α inhibitors might directly inhibit megakaryocytic differentiation, which may lead to the reduction of the PLR.

A recent study revealed that a higher baseline PASI was associated with a higher percentage reduction of the PASI after treatment with biologics if the baseline PASI is ≤ 30 [17]. This phenomenon might contribute to the positive correlation of the baseline PASI with a percentage reduction of the PASI at week 52 in the ADA subgroup (Table 6). The positive correlation of the baseline PASI with a percentage reduction of the PASI by biologics might also be related to the higher achievement rates of PASI 75 and 90 at week 12 or 24 in IFX subgroup with a higher baseline PASI (Table 1) compared to the CZP subgroup (Figure 1a,b). Opposingly, the achievement rate of absolute PASI ≤ 2 in the IFX subgroup was lower than that in the CZP subgroup (Figure 1c), which may be due to the higher baseline PASI in the former. The achievement rates of PASI 75, 90 and absolute PASI ≤ 2 in the IFX subgroup were, however, reduced at week 52 (Figure 1a–c). This might indicate the lack of efficacy in a subpopulation of IFX-treated patients since IFX is a mouse/human chimeric antibody and is associated with a high incidence of anti-drug antibodies, especially of neutralizing antibodies [18].

The presence of scalp lesions might predict a good response to ADA. Though scalp psoriasis belongs to difficult-to-treat types of psoriasis, recent studies revealed that scalp psoriasis showed a good response to ADA [19,20]. Tumor necrosis factor-α might accelerate the pruritus of scalp lesions [21]. It is reported that treatment with ADA reduced the pruritus in scalp psoriasis [20], which might block the scratch-induced Koebner reaction. Patients with scalp lesions may thus be good responders to treatment with ADA. The recent study also reported that patients with scalp lesions showed a higher probability of achieving PASI 75 by ADA and a lower probability of discontinuing ADA compared to those without [22]. In contrast, the presence of genital lesions was associated with a poorer treatment response in all patients and the IFX subgroup at week 52 in this study. Genital psoriasis is known to be highly refractory to topical treatment and is associated with secondary irritation leading to the Koebner reaction and is more commonly associated with obesity and psychological burden [23]. The efficacy of TNF-α inhibitors for genital psoriasis has not been sufficiently evidenced compared to the greater efficacy of the IL-17 inhibitor, ixekizumab [19]. It is also reported that genital psoriasis predicts discontinuation of treatment with ADA due to ineffectiveness [22].

The baseline CRP was negatively correlated with the percentage reduction of the PASI at week 52 of treatment with CZP. This indicates that high systemic inflammation before treatment might predict a poor response to CZP. Based on the present results, when deciding the type of TNF-α inhibitors to treat psoriasis patients, ADA might be applicable to patients with scalp lesions and/or higher baseline PASI scores; CZP might be applicable to patients with a lower baseline CRP; IFX might be applicable to patients without genital lesions. Whether the treatment choice above may be appropriate should be prospectively examined.

The baseline NLR and PLR before treatment were negatively correlated with BMI in this study (Table 2). The findings were opposite to the reported results in general populations, indicating a positive correlation [24]. On the other hand, it is reported that a high NLR is associated with the risk of malnutrition in the geriatric population (>65 years of age) [25], leading to a reduced BMI and indicating that a high NLR is associated with weight loss in cancer patients [26]. The relationship of BMI with the NLR or PLR might differ with age and/or associated diseases. Whether the negative correlation of BMI with the NLR or PLR might be specific to patients with psoriasis or the bias due to the small sample size should further be examined in a larger cohort.

The baseline PLR was negatively correlated with age in this study (Table 2). The results were different from those of Najar Nobari et al., showing a positive correlation [2]. This discrepancy might be due to the lower percentage of females and/or the approx. 10-year older age of the patients in our study. Values of the PLR in females are generally the highest at 40–49 years of age and lower in younger and older individuals, while in males, the PLR values slightly decrease at an age of more than 50 years [27]. The baseline MLR was higher in patients with nail lesions than in those without and was positively correlated with disease duration in this study (Table 2). The results indicate the association of systemic inflammation with nail lesions or longer disease duration in patients with psoriasis. The baseline values of the NLR, PLR, MLR and CRP in patients with psoriasis before treatment should further be analyzed in a larger cohort of patients in relation to BMI, age, sex, disease duration or nail lesions.

This study has several limitations. Firstly, this was a retrospective study with a small number of patients. Especially the sample size of each subgroup treated with IFX, ADA or CZP was small. Further large-scale prospective studies are needed to validate our present findings and to clarify the differences between IFX, ADA and CZP. Secondly, medications for comorbidities or previous treatments for psoriasis, which might influence the baseline PASI or laboratory parameters, were not included in the analysis. Thirdly, we used the conventional method for the measurement of the CRP. The high-sensitivity CRP might be preferable.

## 5. Clinical Relevance of This Study

Testing the CRP and PLR during treatment with TNF-a inhibitors in patients with psoriasis is relevant and useful for monitoring treatment responses to these biologics. The presence of scalp lesions might predict a high treatment response to ADA in patients with psoriasis.

## 6. Conclusions

In conclusion, the NLR, MLR, PLR and CRP values significantly reduced after treatment with TNF-α inhibitors in patients with psoriasis. Percentage reductions of the CRP and PLR were significantly correlated with that of the PASI at week 52 of treatment and a percentage reduction of the CRP was higher in patients with the achievement of PASI 75 than in patients without. The C-reactive protein and PLR might act as biomarkers reflecting a treatment response to TNF-α inhibitors in patients with psoriasis. The presence of scalp lesions might be a predictive factor for a high treatment response to ADA in patients with psoriasis.

## Figures and Tables

**Figure 1 jcm-12-00974-f001:**
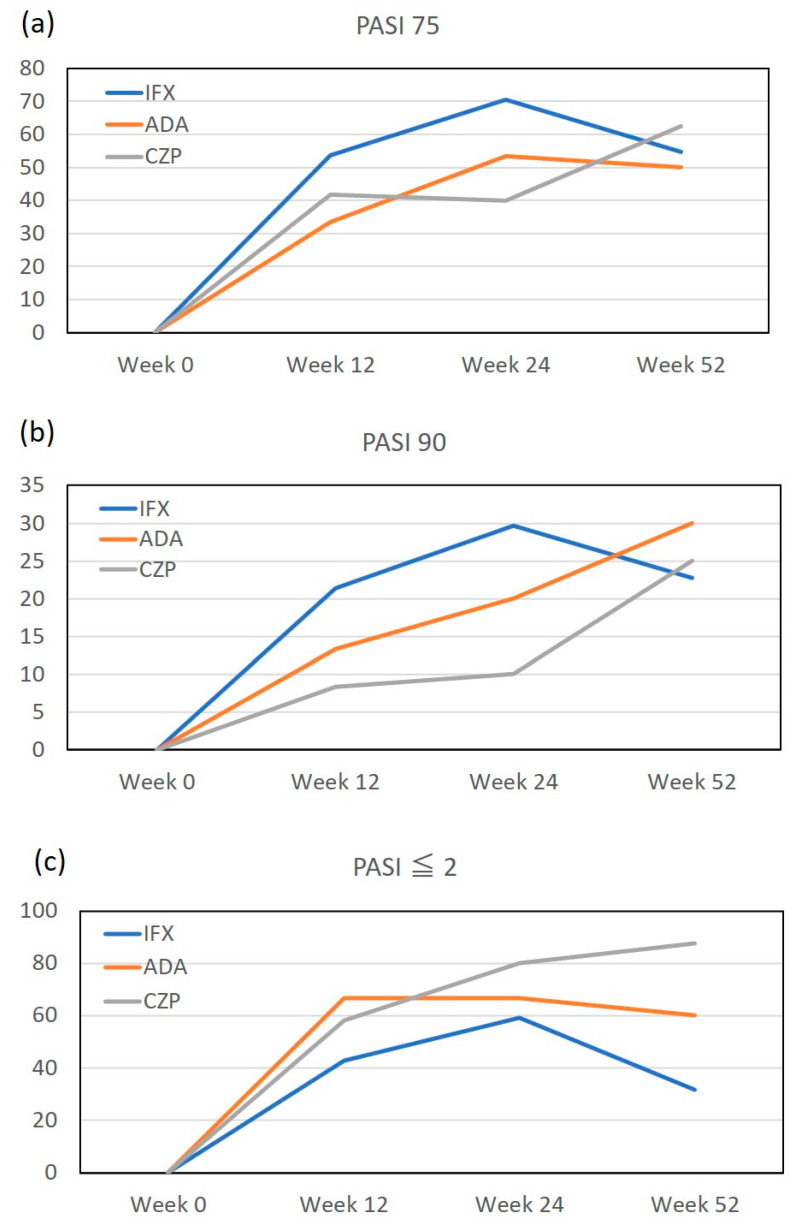
The achievement rates of psoriasis area and severity index (PASI) 75, 90 and absolute PASI ≤ 2 during treatment with individual TNF-α inhibitors in patients with psoriasis. PASI 75 response rate (**a**), PASI 90 response rate (**b**) and PASI ≤ 2 response rate (**c**), respectively.

**Figure 2 jcm-12-00974-f002:**
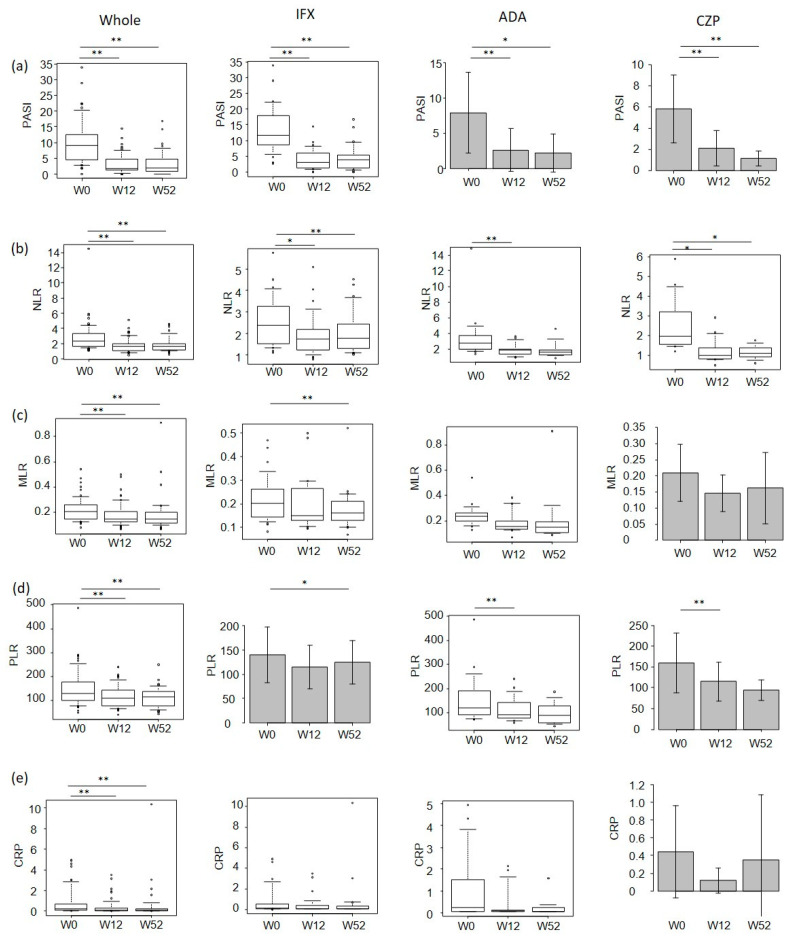
PASI (**a**), NLR (**b**), MLR (**c**), PLR (**d**) and CRP (**e**) values at 0, 12 or 52 weeks of individual TNF-α inhibitor treatments in patients with psoriasis. The ADA and CZP data in (**a**), CZP data in (**c**), IFX and CZP data in (**d**) and CZP data in (**e**) are provided as mean ± standard deviation and analyzed by repeated measure analysis of variance. Otherwise, data are provided as the median (interquartile range) and analyzed by Friedman’s test. * *p* < 0.05; ** *p* < 0.01.

**Table 1 jcm-12-00974-t001:** Baseline characteristics of patients with psoriasis treated with tumor necrosis factor (TNF)-α inhibitors.

Biologic (*n*)	Whole TNF-α Inhibitors (55)	IFX (28)	ADA (15)	CZP (12)	IFX vs. ADA	IFX vs. CZP	ADA vs. CZP
Sex, *n* (%) ^†^							
Male	43 (78.2)	22 (78.6)	12 (80.0)	9 (75.0)	1	1	1
Female	12 (21.8)	6(21.4)	3 (20.0)	3 (25.0)	1	1	1
Age (years) ^‡^	55.33 ± 13.34	56.96 ± 13.91	53.8 ± 13.79	53.42 ± 11.95	1	1	1
Body mass index (kg/m^2^) ^‡^	24.71 ± 4.21	25.48 ± 4.36	22.97 ± 3.67	24.96 ± 4.18	0.21	1	0.69
Disease duration (years) ^§^	16 [8–27]	16.5 [10–27.8]	18.4 ± 9.3	6 [4–26.5]	1	0.4	0.81
Positive bio-switch, *n* (%) ^†^	16 (29.1)	1 (3.6)	10 (66.7)	5 (41.7)	0.000045 **	0.018 *	0.773
Presence of arthritis *n* (%) ^†^	38 (69.1)	15 (53.6)	11(45.8)	12 (100)	0.982	0.011 *	0.318
Presence of scalp lesions, *n* (%) ^†^	41 (74.6)	20 (71.4)	11(45.8)	10 (83.3)	1	1	1
Presence of nail lesions, *n* (%) ^†^	30 (54.6)	15 (53.6)	10 (66.7)	5 (41.7)	1	1	0.77
Presence of genital lesions, *n* (%) ^†^	17 (30.9)	8 (28.6)	3 (20.0)	6 (50.0)	1	1	0.68
Diabetes mellitus, *n* (%) ^†^	8 (14.6)	3 (3.6)	4 (26.7)	1 (8.3)	0.65	1	1
Cardiovascular disease, *n* (%) ^†^	1 (1.8)	1 (3.6)	0 (0)	0 (0)	1	1	1
Current smoking status, ^¶^ *n* (%) ^†^	37 (67.3)	17 (60.7)	12 (80)	8 (66.7)	0.84	1	1
Psoriasis area and severity index ^§^	9 [4.6–12.6]	11.7 [8.6–16.9]	7.94 ± 5.72	5.83 ± 3.21	0.0462 *	0.0022 **	1
Laboratory findings							
Neutrophil-to-lymphocyte ratio ^§^	2.32 [1.61–3.35]	2.54 ± 1.18	2.73 [1.99–3.68]	1.98 [1.58–2.99]	0.6	1	0.65
Monocyte-to-lymphocyte ratio ^§^	0.21 [0.15–0.26]	0.20 [0.14–0.26]	0.23 [0.2–0.26]	0.21 ± 0.088	0.88	1	0.97
Platelet-to-lymphocyte ratio ^§^	129.1 [101.6–178.7]	140.3 ± 50.6	119.4 [92.6–190.1]	159.9 ± 71.5	1	1	1
CRP (mg/dL) ^§^	0.18 [0.09–0.69]	0.165 [0.095–0.57]	0.25 [0.055–1.51]	0.18 [0.15–0.53]	1	1	1

^†^ Differences between groups were analyzed using Fisher’s exact test. ^‡^ Data were provided as mean ± standard deviation, analyzed using one-way analysis of variance. ^§^ Data were provided as the median (interquartile range), analyzed using Kruscal-Wallis test. ^¶^ Four patients whose history of smoking is unknown were excluded from statistical analysis. * *p* < 0.05, ** *p* < 0.01. IFX, infliximab; ADA, adalimumab; CZP, certolizumab pegol; CRP, C-reactive protein.

**Table 2 jcm-12-00974-t002:** The relations between patients’ background factors and baseline values of neutrophil-to-lymphocyte ratio (NLR), monocyte-to-lymphocyte ratio (MLR), platelet-to-lymphocyte ratio (PLR) or C-reactive protein (CRP).

**Sex ^†^**	**Male**	**Female**	** *p* **
*n* (%)	43 (78.2)	12 (21.8)	
NLR	2.32 [1.58–3.20]	2.44 [1.72–4.11]	0.426
MLR	0.208 [0.155–0.259]	0.194 [0.125–0.281]	0.636
PLR	129.1 [93.0–177.8]	141.2 [120.6–272.5]	0.178
CRP (mg/dL)	0.18 [0.08–0.60]	0.32 [0.11–0.83]	0.547
Bio-switch ^†^	Absence	Presence	*p*
*n* (%)	39 (70.9)	16 (29.1)	
NLR	2.32 [1.56–3.24]	2.76 [1.77–3.78]	0.29
MLR	0.196 [0.142–0.257]	0.239 [0.171–0.283]	0.0735
PLR	128.3 [98.4–165.8]	161.0 [111.4–200.8]	0.265
CRP (mg/dL)	0.180 [0.100–0.675]	0.215 [0.058–0.655]	0.985
Arthritis ^†^	Absence	Presence	*p*
*n* (%)	17 (30.9)	38 (69.1)	
NLR	2.56 [1.68–3.23]	2.25 [1.60–3.38]	0.978
MLR	0.215 [0.146–0.253]	0.204 [0.150–0.271]	0.935
PLR	128.3 [104.2–155.2]	132.8 [93.3–185.0]	0.557
CRP (mg/dL)	0.22 [0.10–0.62]	0.18 [0.085–0.713]	0.708
Scalp lesions ^†^	Absence	Presence	*p*
*n* (%) ^‡^	13 (23.6)	41 (74.5)	
NLR	1.968 [1.726–3.299]	2.537 [1.621–3.407]	0.795
MLR	0.167 [0.141–0.196]	0.222 [0.163–0.273]	0.0513
PLR	128.3 [113.7–193.1]	134.1 [99.0–178.5]	0.81
CRP (mg/dL)	0.18 [0.05–0.62]	0.18 [0.10–0.75]	0.641
Nail lesions ^†^	Absence	Presence	*p*
*n* (%) ^‡^	23 (41.8)	30 (54.5)	
NLR	2.32 [1.67–3.53]	2.43 [1.64–3.49]	0.979
MLR	0.186 [0.144–0.212]	0.255 [0.158–0.296]	0.0165 *
PLR	134.1 [106.5–190.1]	128.7 [92.2–178.2]	0.587
CRP (mg/dL)	0.16 [0.075–0.595]	0.205 [0.11–1.015]	0.31
Genital lesions ^†^	Absence	Presence	*p*
*n* (%) ^‡^	31 (56.3)	17 (25.5)	
NLR	2.32 [1.77–3.06]	3.08 [1.59–4.34]	0.494
MLR	0.215 [0.155–0.268]	0.186 [0.146–0.256]	0.782
PLR	136.5 [106.5–178.7]	129.1 [113.7–193.1]	0.881
CRP (mg/dL)	0.16 [0.055–0.855]	0.46 [0.11–0.78]	0.235
Diabetes mellitus ^†^	Absence	Presence	*p*
*n* (%)	47 (85.5)	8 (14.5)	
NLR	2.32 [1.65–3.35]	2.37 [1.54–3.19]	0.716
MLR	0.210 [0.152–0.265]	0.197 [0.127–0.226]	0.378
PLR	137.1 [106.7–182.9]	110.5 [75.7–134.7]	0.103
CRP (mg/dL)	0.18 [0.08–0.685]	0.135 [0.11–0.585]	0.729
Cardiovascular disease ^†^	Absence	Presence	*p*
*n* (%)	54 (98.2)	1 (1.8)	
NLR	2.38 [1.64–3.38]	1.51	0.299
MLR	0.209 [0.150–0.263]	0.081	0.095
PLR	131.6 [104.3–131.6]	50.6	0.095
CRP (mg/dL)	0.18 [0.10–0.7175]	0.08	0.344
Current smoking status ^†^	Absence	Presence	*p*
*n* (%) ^‡^	14 (25.5)	37 (67.3)	
NLR	2.25 [1.64–2.53]	2.70 [1.62–3.95]	0.33
MLR	0.179 [0.136–0.232]	0.215 [0.156–0.277]	0.144
PLR	127.6 [113.5–183.7]	134.1 [97.8–178.6]	0.654
CRP (mg/dL)	0.12 [0.0575–0.5425]	0.18 [0.11–0.76]	0.15
Past tuberculosis ^†^	Absence	Presence	*p*
*n* (%)	51 (92.7)	4 (7.3)	
NLR	2.54 [1.61–3.49]	2.15 [1.80–2.35]	0.339
MLR	0.208 [0.145–0.262]	0.225 [0.181–0.264]	0.685
PLR	134.1 [104.4–182.9]	104.7 [84.9–127.5]	0.212
CRP (mg/dL)	0.18 [0.08–0.61]	0.64 [0.4325–0.8375]	0.284
Correlations with age ^§^		*rho*	*p*
NLR		−0.207	0.13
MLR		−0.204	0.135
PLR		−0.43	0.00104 **
CRP (mg/dL)		−0.0544	0.693
Correlations with BMI ^§^		*rho*	*p*
NLR		−0.343	0.0111 *
MLR		−0.248	0.0701
PLR		−0.379	0.00468 **
CRP (mg/dL)		−0.121	0.363
Correlations with disease duration ^§^		*rho*	*p*
NLR		−0.0666	0.629
MLR		0.269	0.0473 *
PLR		0.105	0.446
CRP (mg/dL)		−0.144	0.294

^†^ Data were provided using median (interquartile range) and analyzed using Mann-Whitney U test. ^‡^ The patients whose status is unknown were excluded from statistical analyses; one patient for scalp lesions, two patients for nail lesions, seven patients for genital lesions, four patients for current smoking status. ^§^ Correlations between variables were examined using Spearman’s correlation coefficients. * *p* < 0.05, ** *p* < 0.01.

**Table 3 jcm-12-00974-t003:** Correlations between individual laboratory parameters.

		Week 0		Week 12		Week 52	
Biologic (*n*)	Parameters	*rho*	*p*	*rho*	*p*	*rho*	*p*
Whole TNF-α inhibitors (55)	NLR vs. MLR	0.64	0.000000324 **	0.596	0.00000154 **	0.732	4.08 × 10^−7^ **
	NLR vs. PLR	0.725	< 2.2 × 10^−16^ **	0.624	0.0000007 **	0.485	0.0017 **
	NLR vs. CRP	0.504	0.000086 **	0.539	0.0000219 **	0.368	0.0194 **
	MLR vs. PLR	0.555	0.0000155 **	0.485	0.000172 **	0.479	0.00197 **
	MLR vs. CRP	0.293	0.0298 *	0.305	0.0234 *	0.457	0.00307 **
	PLR vs. CRP	0.236	0.0833	0.155	0.258	0.274	0.087
IFX (28)	NLR vs. MLR	0.758	0.00000632 **	0.644	0.000302 **	0.698	0.000431 **
	NLR vs. PLR	0.84	0.00000135 **	0.69	0.0000741 **	0.698	0.000431 **
	NLR vs. CRP	0.463	0.0132 *	0.536	0.00325 **	0.478	0.0246 *
	MLR vs. PLR	0.694	0.0000653 **	0.687	0.0000825 **	0.713	0.000288 **
	MLR vs. CRP	0.454	0.0151 *	0.349	0.0691	0.638	0.0014 **
	PLR vs. CRP	0.406	0.0321*	0.24	0.219	0.287	0.195
ADA (15)	NLR vs. MLR	0.389	0.152	0.482	0.0711	−0.212	0.56
	NLR vs. PLR	0.711	0.00406 **	0.654	0.01 *	0.77	0.0137 *
	NLR vs. CRP	0.605	0.0168 *	0.569	0.0268 *	0.382	0.276
	MLR vs. PLR	0.511	0.0543	0.325	0.237	0.0545	0.892
	MLR vs. CRP	0.319	0.247	0.131	0.641	0.273	0.445
	PLR vs. CRP	0.317	0.249	0.161	0.567	0.287	0.422
CZP (12)	NLR vs. MLR	0.35	0.266	0.515	0.0867	0.762	0.0368 *
	NLR vs. PLR	0.65	0.0259 *	0.573	0.0555	0.738	0.0458 *
	NLR vs. CRP	0.472	0.121	0.269	0.399	−0.0479	0.91
	MLR vs. PLR	0.476	0.121	0.193	0.549	0.31	0.462
	MLR vs. CRP	−0.0916	0.777	0.0549	0.866	0.0719	0.866
	PLR vs. CRP	−0.0352	0.913	−0.113	0.726	0.18	0.67

Correlations between variables were examined using Spearman’s correlation coefficients. * *p* < 0.05, ** *p* < 0.01; TNF-α, tumor necrosis factor-α; IFX, infliximab; ADA, adalimumab; CZP, certolizumab pegol; NLR, neutrophil-to-lymphocyte ratio; MLR, monocyte-to-lymphocyte ratio; PLR, platelet-to-lymphocyte ratio; CRP, C-reactive protein.

**Table 4 jcm-12-00974-t004:** Correlations between percentage reduction of psoriasis area and severity index versus those of neutrophil-to-lymphocyte ratio (NLR), monocyte-to-lymphocyte ratio (MLR), platelet-to-lymphocyte ratio (PLR) or C-reactive protein (CRP).

		Week 12		Week 52	
Biologic (*n*)	Laboratory Parameters	*rho*	*p*	*rho*	*p*
Whole TNF-α inhibitors (55)	NLR	0.0795	0.564	0.24	0.135
	MLR	0.0406	0.768	0.251	0.119
	PLR	0.0287	0.835	0.356	0.0243 *
	CRP	0.222	0.103	0.418	0.00725 **
IFX (28)	NLR	0.0342	0.863	0.158	0.482
	MLR	0.00904	0.964	0.219	0.327
	PLR	0.102	0.606	0.218	0.33
	CRP	0.268	0.168	0.466	0.0288 *
ADA (15)	NLR	0.192	0.512	0.317	0.41
	MLR	0.258	0.374	−0.05	0.912
	PLR	0.0705	0.811	0.4	0.291
	CRP	0.288	0.319	0.617	0.0857
CZP (12)	NLR	0.273	0.39	−0.599	0.117
	MLR	−0.035	0.914	0.311	0.453
	PLR	−0.396	0.203	−0.168	0.691
	CRP	0.378	0.226	−0.247	0.555

Correlations between variables were examined using Spearman’s correlation coefficients. * *p* < 0.05, ** *p* < 0.01; TNF-α, tumor necrosis factor-α; IFX, infliximab; ADA, adalimumab; CZP, certolizumab pegol.

**Table 5 jcm-12-00974-t005:** Percentage reductions of laboratory parameters in high responders versus low responders during treatment with tumor necrosis factor (TNF)-a inhibitors.

		Week 12			Week 52		
		Percent Reduction of PASI			Percent Reduction of PASI		
Biologics	Laboratory Parameters	<75%	≥75%	*p*	<75%	≥75%	*p*
Whole TNF-α inhibitors	*n*	28	27		18	22	
	NLR	34 [13–49]	40 [16–60]	0.457	27 [6–49]	39 [19–52]	0.325
	MLR	17 [–7–33]	29 [–12–42]	0.336	7 [–4–38]	26 [13–37]	0.251
	PLR	22 [–5–30]	20 [10–42]	0.553	22 [–11–31]	27 [6–43]	0.163
	CRP	27 [–10–62]	55 [7–93]	0.0676	15 [–19–68]	63 [51–90]	0.0106 *
IFX	*n*	11	17		10	12	
	NLR	30 [−1–35]	24 [11–44]	0.677	22 [6–27]	19 [11–40]	0.821
	MLR	7 [−17–28]	25 [−16–41]	0.611	15 [2–39]	26 [8–32]	0.872
	PLR	4 [−6–29]	20 [−11–42]	0.547	13 [−12–25]	11 [1–44]	0.346
	CRP	−19 [−65–29]	38 [13–92]	0.0407 *	8 [−34–43]	59 [35–81]	0.0697
ADA	*n*	10	5		5	5	
	NLR	29 [8–43]	61 [18–64]	0.44	33 [−21–43]	61 [52–68]	0.0952
	MLR	21 [−8–34]	32 [13–57]	0.594	3 [−9–33]	32 [30–41]	0.421
	PLR	22 [−2–27]	13 [13–44]	0.44	28 [−21–32]	42 [39–50]	0.0556
	CRP	32 [4–56]	87 [0–94]	0.58	17 [0–71]	78 [77–99]	0.0159 *
CZP	*n*	7	5		3	5	
	NLR	49 [39–50]	58 [49–70]	0.268	50 [49–60]	44 [37–49]	0.571
	MLR	17 [1–37]	32 [30–39]	0.343	−5 [−65–17]	27 [18–27]	0.25
	PLR	30 [22–38]	22 [21–28]	0.53	33 [28–43]	26 [17–28]	0.25
	CRP	71 [50–91]	89 [28–90]	0.568	72 [26–79]	60 [56–60]	1

PASI, psoriasis area and severity index; IFX, infliximab; ADA, adalimumab; CZP, certolizumab pegol; NLR, neutrophil-to-lymphocyte ratio; MLR, monocyte-to-lymphocyte ratio; PLR, platelet-to-lymphocyte ratio; CRP, C-reactive protein. Data are provided as the median (interquartile range), analyzed using Mann-Whitney U test. * *p* < 0.05.

**Table 6 jcm-12-00974-t006:** The correlations between percentage reduction of psoriasis area and severity index (PASI) at week 52 versus age, body mass index (BMI), disease duration, baseline values of PASI or laboratory parameters.

	Whole TNF-α Inhibitors (55)		IFX (28)		ADA (15)		CZP (12)	
Background Factors	*rho*	*p*	*rho*	*p*	*rho*	*p*	*rho*	*p*
Age	−0.0433	0.791	0.032	0.888	−0.164	0.657	−0.192	0.649
BMI	−0.00324	0.984	0.188	0.402	−0.31	0.417	0.0958	0.821
Disease duration	−0.197	0.224	−0.299	0.177	−0.178	0.623	−0.0783	0.854
Baseline PASI	0.248	0.122	0.225	0.315	0.648	0.049 *	0.614	0.105
Baseline CRP	0.195	0.229	0.164	0.467	0.584	0.0765	−0.735	0.0378 *
Baseline NLR	−0.104	0.522	−0.312	0.158	0.164	0.657	−0.0599	0.888
Baseline MLR	−0.1	0.537	−0.153	0.496	0.00606	1	0	1
Baseline PLR	−0.0401	0.806	−0.167	0.459	0.0182	0.973	0.204	0.629

Correlations between variables were examined using Spearman’s correlation coefficients. TNF-α, tumor necrosis factor-α; IFX, infliximab; ADA, adalimumab; CZP, certolizumab pegol; NLR, neutrophil-to-lymphocyte ratio; MLR, monocyte-to-lymphocyte ratio; PLR, platelet-to-lymphocyte ratio; CRP, C-reactive protein; * *p* < 0.05.

**Table 7 jcm-12-00974-t007:** The relationship between percentage reduction of psoriasis area and severity index (PASI) at week 52 versus sex, presence or absence of arthritis, scalp or genital lesions.

		Sex			Arthritis			Scalp Lesions			Genital Lesions		
Biologics (*n*)		Male	Female	*p*	Absence	Presence	*p*	Absence	Presence	*p*	Absence	Presence	*p*
Whole TNF-α inhibitors (55) ^†^	*n* (%)	43 (78.2)	12 (21.8)		17 (30.9)	38 (69.1)		13 (23.6)	41 (74.5)		31 (56.3)	17 (30.9)	
	Percent reduction of PASI	74 [43–93]	83 [76–87]	0.327	69 [36–84]	76 [58–90]	0.345	75 [14–86]	76 [57–89]	0.44	83 [61–94]	60 [44–81]	0.0378 *
IFX (28) ^†^	*n* (%)	6 (21.4)	22 (78.6)		13 (46.4)	15 (53.6)		7 (25.9)	20 (71.4)		15 (53.6)	8 (28.6)	
	Percent reduction of PASI	75 [46–91]	72 [62–81]	1	68 [46–81]	76 [62–87]	0.307	78 [75–95]	65 [49–82]	0.319	82 [65–95]	56 [40–75]	0.0383 *
ADA (15) ^‡^	*n* (%)	12 (80.0)	3 (20.0)		4 (26.7)	11 (73.3)		4 (26.7)	11 (73.3)		10 (66.7)	3 (20)	
	Percent reduction of PASI	48.0 ± 49.6	87.3 ± 1.8	0.316	33.3 ± 94.3	61.5 ± 36.9	0.479	−4.9 ± 26.3	82.0 ± 20.2	0.00043 **	61.8 ± 41.4	26.4 ± 84.4	0.404
CZP (12) ^‡^	*n* (%)	9 (75.0)	3 (25.0)		0 (0)	12 (100)		2 (16.7)	10 (83.3)		6 (50)	6 (50)	
	Percent reduction of PASI	72.5 ± 17.2	88.0 ± 6.6	0.278	Not done			75.5 ± 24.2	76.6 ± 16.1	0.938	81.4 ± 19.1	71.3 ± 14.0	0.422

TNF-α, tumor necrosis factor-α; IFX, infliximab; ADA, adalimumab; CZP, certolizumab pegol † Data are provided as the median (interquartile range), analyzed using Mann-Whitney U test. ‡ Data are provided as mean ± standard deviation, analyzed using Student’s *t*-test. * *p* < 0.05, ** *p* < 0.01.

**Table 8 jcm-12-00974-t008:** The relationship between percentage reduction of psoriasis area and severity index (PASI) at week 52 versus presence or absence of nail lesions, diabetes mellitus, current smoking status or bio-switch.

		Nail Lesions			Diabetes Mellitus			Current Smoking			Bio-Switch		
Biologics (*n*)		Absence	Presence	*p*	Absence	Presence	*p*	Absence	Presence	*p*	Absence	Presence	*p*
Whole TNF-α inhibitors (55) ^†^	*n* (%)	23 (41.8)	30 (54.6)		47 (85.5)	8 (14.6)		14 (25.5)	37 (67.3)		39 (70.9)	16 (29.1)	
	Percent reduction of PASI	81 [53–90]	75 [56–89]	0.746	76 [56–88]	74 [55–91]	0.805	76 [51–83]	76 [57–92]	0.797	76 [56–90]	74 [45–88]	0.755
IFX (28) ^†^	*n* (%)	11 (39.3)	15 (53.6)		25 (89.3)	3 (10.7)		8 (28.6)	17 (60.7)		27 (96.4)	1 (3.6)	
	Percent reduction of PASI	80 [51–83]	70 [53–84]	0.722	76 [53–85]	58 [55–62]	0.458	80 [60–88]	75 [56–83]	0.663	76 [51–83]	75 [75,75]	1
ADA (15) ^‡^	*n* (%)	5 (33.3)	10 (66. 7)		11 (73.3)	4 (26.7)		2 (13.3)	12 (80.0)		5 (33.3)	10 (66.7)	
	Percent reduction of PASI	28.4 ± 67.2	67.7 ± 35.2	0.245	52.2 ± 46.8	64.5 ± 55.9	0.727	18.5 ± 0.0	60.0 ± 47.6	0.432	64.5 ± 55.9	52.2 ± 46.8	0.727
CZP (12) ^‡^	*n* (%)	7 (58.3)	5 (41.7)		11 (91.7)	1 (8.3)		4 (33.3)	8 (66.7)		7 (58.3)	5 (41.7)	
	Percent reduction of PASI	79.1 ± 17.7	68.0 ± 11.3	0.449	81.4 ± 19.1	71.3 ± 14.0	0.422	72.2 ± 13.5	78.9 ± 18.9	0.615	75.1 ± 14.5	80.0 ± 28.3	0.745

TNF-α, tumor necrosis factor-α; IFX, infliximab; ADA, adalimumab; CZP, certolizumab pegol † Data are provided as the median (interquartile range), analyzed using Mann-Whitney U test. ‡ Data are provided as mean ± standard deviation, analyzed using Student’s *t*-test.

**Table 9 jcm-12-00974-t009:** The predictive factors for percentage reduction of psoriasis area and severity index (PASI) at week 52 analyzed using linear multivariate regression analysis.

Biologic (*n*)	Whole TNF-α Inhibitors (55)				IFX (28)				ADA (15)				CZP (12)			
	β Coefficient	Standard Error	*t*	*p*	β Coefficient	Standard Error	*t*	*p*	β Coefficient	Standard Error	*t*	*p*	β Coefficient	Standard Error	*t*	*p*
(Intercept)	0.152	0.488	0.311	0.758	0.211	0.799	0.264	0.796	742.4	1132.1	0.657	0.559	−937.3	411.1	−2.2801	0.107
Age	−0.001	0.004	−0.321	0.750	−0.004	0.006	−0.791	0.442	0.0084	0.0047	1.796	0.170	0.00297	0.00397	0.7467	0.509
Sex (Male = 1, female = 2)	0.245	0.157	1.558	0.129	0.158	0.252	0.627	0.541	0.0977	0.4132	0.2364	0.828	0.32100	0.14414	2.2271	0.112
BMI (kg/m^2^)	0.014	0.014	1.0378	0.307	0.024	0.019	1.247	0.233	3.37 × 10^−7^	0.00000	0.6563	0.558	−4.2 × 10^−7^	0.000	−2.2815	0.107
Baseline PASI	NA								0.0343	0.0138	2.4885	0.089	NA			
Baseline CRP (mg/dL)	NA								NA				−0.1563	0.090448	−1.7284	0.182
Scalp lesions	NA								0.8279	0.156	5.2933	0.013 *	NA			
Genital lesions	−0.230	0.121	−1.895	0.067	−0.285	0.165	−1.730	0.106	NA				NA			

TNF-α, tumor necrosis factor-α; IFX, infliximab; ADA, adalimumab; CZP, certolizumab pegol; BMI, body mass index; CRP, C-reactive protein; NA, not applicable; * *p* < 0.05.

## Data Availability

Not applicable.

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
