# Peer review of "Biomarkers and Predictive Factors for Treatment Response to Tumor Necrosis Factor-α Inhibitors in Patients with Psoriasis"

_jcm, 2023, doi:10.3390/jcm12030974_

Round 1
Reviewer 1 Report
Point 1: Line 9: In the abstract, you must include an introduction (1 or 2 sentences) that summarizes your topic and highlights the purpose of the study. Also, the type of study
Point 2: references 1 and 2 are the same
Point 3: line 44-46 make it fluent: It has been reported that baseline NLR and PLR before treatment may predict TNF- inhibitor treatment outcomes in patients with rheumatoid arthritis; higher baseline NLR and PLR were associated with non-response to TNF- inhibitors at week 12.
Point 4: line 51 add Biomarkers definition according to the National Institutes of Health Biomarkers Definitions Working Group,
Point 5: study design and data collection: Ideally, a section describing Standard treatment schemes should be included.
Pint 6: If you could highlight the most important findings in the table
Point 7: 317 add reference
Point 8: Add a section on the relevance of the study
Reviewer 2 Report
The manuscript is very interesting and provides useful and simple to use methods to predict therapeutic response to anti-TNF-a.
The main question addressed by the research is to define patient's background and biomarkers to predict efficacy and response to biological treatment with anti-TNF-a in psoriasis.
Is it relevant and useful as those biomarkers proposed are routinely used or not very difficult to obtain
The topic is original and pragmatic especially now when sanitary costs are quite well considered: this study really helps clinician choosing the most suitable biologic for each patient.
The research adds more insights in this topic and new evidence especially on regards of scalp psoriasis and CPR use.
The paper is well written and the text is clear to read and understand.
Then, the conclusions are consistent with the evidence and the arguments presented.
